# A Comparison of the Effect of Cellulose Nanocrystals (CNCs) and Polyethylene Glycol (PEG) as Additives in Ultrafiltration Membranes (PES-UF): Characterization and Performance

**DOI:** 10.3390/polym15122636

**Published:** 2023-06-09

**Authors:** Amos Adeniyi, Gerald Oke Odo, Danae Gonzalez-Ortiz, Celine Pochat-Bohatier, Sandrine Mbakop, Maurice Stephen Onyango

**Affiliations:** 1Department of Chemical, Metallurgical and Materials Engineering, Tshwane University of Technology, Pretoria 0183, South Africa; geraldokeyodo@yahoo.com (G.O.O.); sandrine.mbakop88@gmail.com (S.M.); 2Water for Rural Communities (WARUC), Pretoria 0002, South Africa; 3Institut Européen des Membranes, IEM UMR-5635, Université de Montpellier, ENSCM, CNRS Place Eugène Bataillon, CEDEX 5, 34095 Montpellier, France; danae.gonzales-ortiz@umontpellier.fr (D.G.-O.); celine.pochat@umontpellier.fr (C.P.-B.)

**Keywords:** ultrafiltration, PEG, CNC, COD, turbidity, morphology

## Abstract

This work demonstrated the potential of CNC as a substitute for PEG as an additive in ultrafiltration membrane fabrication. Two sets of modified membranes were fabricated using the phase inversion technique, with polyethersulfone (PES) as the base polymer and 1-N-methyl-2 pyrrolidone (NMP) as the solvent. The first set was fabricated with 0.075 wt% CNC, while the second set was fabricated with 2 wt% PEG. All membranes were characterized using SEM, EDX, FTIR, and contact angle measurements. The SEM images were analyzed for surface characteristics using WSxM 5.0 Develop 9.1 software. The membranes were tested, characterized, and compared for their performance in treating both synthetic and real restaurant wastewater. Both membranes exhibited improved hydrophilicity, morphology, pore structure, and roughness. Both membranes also exhibited similar water flux for real and synthetic polluted water. However, the membrane prepared with CNC gave higher turbidity removal and COD removal when raw restaurant water was treated. The membrane compared well with the UF membrane containing 2 wt% PEG in terms of morphology and performance when synthetic turbid water and raw restaurant water were treated.

## 1. Introduction

The global water crisis, which includes water scarcity, water pollution, water contamination, and ecosystem degradation, has become a big and general challenge in society [1]. The ability of the ecosystem to provide fresh water supplies is becoming increasingly compromised. Freshwater becomes polluted because of industrial, agricultural, and domestic activities [2]. Domestic wastewater includes yellow water, black water, brown water, and grey water [3]. The environmental and health impacts of these wastewaters have become a big problem and worry in society. In a world with an increase in water use, water treatment, reuse, and recycling are becoming increasingly important [4]. Techniques that apply to a wide range of water matrices are membrane processes [5].

Membrane filtration processes are fast becoming the preferred technique for water and wastewater treatment [6]. This is due to the low cost of wastewater treatment and the easy installation. The membrane process is eco-friendly, energy consumption is lower than other technologies, and it has a 90–95% recovery rate of water [7]. Membrane filtration processes have wide industrial and commercial applications. Many industries, such as food and beverage production and oil and gas, use membrane processes for separating solids from solutions. For example, the membrane filtration process provides a highly desirable method, mostly for treating wastewater, due to its operation efficiency [8,9,10]. The efficiency of the membrane in separation is generally based on the size of the pores and the wetting properties of the membrane [11]. Membrane pore sizes act as a physical barrier to large particles in the water solution, allowing water to pass through under pressure while obstructing pollutants in the water that are larger than the membrane pore. Generally, considering the pore size and separation mechanism, pressure-driven membrane filtration is divided into microfiltration (MF), ultrafiltration (UF), nanofiltration (NF), and reverse osmosis (RO). These can further be classified into low pressure and high pressure, depending on the driving force employed in water treatment. Reverse osmosis and the nanofiltration method are high-pressure processes, while microfiltration and ultrafiltration are known as low-pressure membrane processes. All these membranes are liable to fouling. This fouling mostly occurs as a result of different interactions between a component in the feed water solution and the membrane surface. The major significant advantages of the membrane separation process include operation at room temperature without phase change, compact, easy installation, simplicity in operation, and excellent water quality [12].

Ultrafiltration (UF) is a low-pressure membrane filtration process that can remove suspended solids and bacteria from water. The operating pressure of the UF membrane is normally less than 100 kPa (14.5 psi). The pore size is approximately 0.002 to 0.1 microns, and the molecular weight cut-off (MWCO) is approximately 10,000 to 100,000 Daltons [13]. It is used to remove contaminants, such as bacteria, protozoa, and some viruses, from the water. In UF, the only variables that can affect contaminant removal are pore size and contaminant size; however, both the pore size and hydrophobicity of the membrane have an impact on water recovery. PEG is normally added to influence the membrane pore size and improve its water affinity. Therefore, the use of naturally synthesized additives that could improve the pore size and water affinity of the membrane has the potential to improve the performance of the membrane [14].

UF membranes are porous and are commonly fabricated using the nonsolvent-induced phase separation method [15]. In this method, cast polymer dispersion is immersed in a nonsolvent bath, leading to the formation and growth of polymer-rich and polymer-lean phases within the cast film. Polymers, such as polyethersulfone (PES), polysulfone, polyacrylonitrile, and polypropylene, are commonly used for the preparation of UF membranes [16]. These polymers are added to a solvent that is usually N-methyl-2-pyrrolidone (NMP) or dimethylformamide (DMF). Hydrophilic polymers, such as cellulose acetate, chitosan, PVP, and PEG, are common additives in casting solutions to obtain improved hydrophilic properties [17]. Polyethylene glycol (PEG) acts as a pore former and is known to influence the pore density on the membrane surface [18]. Cellulose nanocrystals (CNCs) are natural nanomaterials primarily derived from naturally occurring cellulose fiber [19]. CNCs have received significant interest in wide application due to their chemical and mechanical properties [19]. CNCs have been reported as one of the emerging materials for wastewater treatment because they are non-toxic, renewable, biodegradable, and possess high specific strength [16,20]. The CNC is hydrophilic, sustainable, and environmentally friendly in most applications.

In one of our previous works, we reported the performance of a UF membrane fabricated with CNCs [21]. CNC concentration was varied as an additive at 0.05%, 0.075%, 0.1%, and 0.15%. While 0.05% of CNC was too low for a significant effect on the PES-UF membrane, a concentration higher than 0.075% was not homogenous, resulting in the formation of agglomerates (Figure 1). There was no significant change when 0.05% CNC was added; however, the addition of 0.075% CNC brought a change to the morphology of the membrane. The surface roughness increased with the increase in CNC concentration. The degree of formation of agglomerates was higher for CNC0.1, and this was clearly reflected in the roughness of the membrane, leading to a membrane with the highest roughness. The EDS results indicated the presence of CNCs in the membrane material. The results showed a gradual increase in the mass concentration of the oxygen as the percentage of CNCs increased: PES (20.13% oxygen), CNC0.05 (20.42% oxygen), CNC0.075 (20.92% oxygen), CNC0.1 (23.36% oxygen), and CNC0.15 (26.95% oxygen). CNC0.075 had the lowest contact angle (69°) and was the only CNC membrane with a lower contact angle than the PES membrane (71.4°). This was due to the homogeneity of the membrane with the CNC content. It was found that average water flux, turbidity, and COD removal from restaurant wastewater were highest with PES membranes containing 0.075 wt% CNC. This was because of the lower contact angle and lower pore size distribution, but this work seeks to compare the use of CNCs with PEG as additives to PES membranes to show that CNCs are a suitable substitute for PEG and other additives.

## 2. Materials and Methods

### 2.1. Materials

Polyethersulfone (PES) was obtained from Solvay, while the non-woven support was obtained from Kavon Filters. 1-N-methyl-2 pyrrolidone (NMP) (>99%) and PEG4000 were obtained from Sigma Aldrich. Commercial cellulose nanocrystal (CNC) powder was obtained from CelluForce, Canada. The CNCs were produced by hydrolyzing bleached softwood kraft wood pulp in sulfuric acid, followed by neutralization with sodium hydroxide. The supplier stated that the CNCs had an overall particle size range of 1 to 50 nm, a length range of 44 to 108 nm, a gram molecular weight range of 14,700 to 27,850, and a diameter range of 2.3 to 4.5 nm. Restaurant wastewater was obtained from a city restaurant in South Africa. Synthetic turbid water was prepared with deionized water and zeolite.

### 2.2. Membrane Fabrication

The membrane was fabricated using the phase inversion technique. The separation of an initially homogeneous mixture into two separate phases, each containing a polymer, a solvent, and additional additives, is known as phase inversion [22]. The solid phase, which is the polymer-rich phase, will give rise to the membrane matrix, whereas the solvent-rich liquid phase, also known as the polymer-lean phase, will lead to the formation of the membrane pores. A polymer solution was prepared with polyethersulfone (PES) (18 wt%) in 1-N-methyl-2 pyrrolidone (NMP) as solvent. The PES was dried in an oven for 24 h before use to eliminate any water that may be present. The solution was stirred using a magnetic stirrer for 24 h before use. The stirring was conducted at 40 °C until the solution was clear and homogeneous. The polymer solution was then used to cast the membrane on a non-woven support using an automatic casting machine. The non-woven support was fixed to a glass plate using cello tape. The thickness was set at 250 µm using the micrometer gauge on the casting knife in a knife film applicator Elcometer, 3580 (Elcometer, Manchester, UK)). The polymer was then immersed in a bowl of water at room temperature. The procedure was carried out for two separate polymer solutions containing cellulose nanocrystals (CNCs) at 0.075 wt% and PEG at 2 wt%. Thus, 0.075 wt% was chosen for CNCs based on our previous work [21], while 2 wt% of PEG is common in the literature [23,24].

### 2.3. Membrane Characterization

The membranes were characterized using a contact angle analyzer, scanning electron microscopy (SEM), Fourier-transform infrared (FTIR), and energy-dispersive X-ray (EDX). The procedure is well-documented in the literature [25]. The SEM images were analyzed for roughness, pore size distribution, and morphology using WSxM 5.0 Develop 9.1 Software [26]. The procedure is as follows: open the software, select the image, go to display, and choose 3D. Click roughness analysis to display roughness analysis results. The 2D image was profiled to obtain the pore size distribution. The procedure is as follows: Go back to the 2D image, select the image, go to process, select profile, left-click the edge of the image to create the profile, and right-click the edge of the image to end the process. Then, select the profile and change it to a histogram.

### 2.4. Analysis

Separation performance was conducted in a dead-end filtration cell, Steritech HP 4750 pressure membrane test cell (Sligo, Rathquarter, Ireland). The active membrane area in the cell was 14.6 cm^2^. The membranes were tested for pure water flux and turbidity removal with synthetic turbid water. The pressure was varied from 0.2 bar to 1.0 bar, while the initial turbidity was varied from 80 NTU to 140 NTU. The membranes were also used for the treatment of raw restaurant wastewater. Chemical oxygen demand (COD) was analyzed using a spectrophotometer, Hach DR6000 UV-VIS (Loveland, CO, USA), while turbidity was analyzed with a turbidity meter (Hach 2100Q Portable Turbidimeter).

The water flux was calculated using Equation (1), while turbidity and COD were calculated using Equation (2).
(1)$$J_{w}=\frac{V}{At}$$
where $J_{w}$ (L/m^2^/h) and *V* (L) are the water flux and the permeate volume, respectively. The active membrane area *A* is measured in m^2^, while the filtration time *t* is in h.
(2)$$R=\frac{\left(C_{f}-C_{p}\right)}{C_{f}}\times 100$$
where *R* is the rejection in %, *C_f_* is the initial turbidity or COD in the feed, and *C_p_* is the final turbidity or COD in the permeate.

## 3. Results and Discussion

In this section, the PES-UF membrane with 0.075 wt% CNC (CNC0.075) is compared with the PES-UF membrane with 2 wt% PEG (PEG). PEG is an established additive in UF membranes to improve pore structure and hydrophilicity [27].

### 3.1. Comparing the Membrane Characteristics

Figure 2 shows the spectra of the CNC0.075 and PEG membranes. Both membranes show similar peaks that are usually observed for UF membranes fabricated with PES. Among the notable peaks are the peaks 1487 cm^−1^ and 1581 cm^−1^, which indicate C-C bond stretching and the benzene ring stretching, respectively. The PEG shows a strong absorption peak at 1700 cm^−1^ that was not observed in CNC0.075. This indicates a strong presence of O-H bending in the membrane [28]. However, both membranes show a strong hydrogen-bonded OH stretching at around 3100 cm^−1^ that may have an impact on the hydrophilicity [29].

The EDX spectra for both PEG and CNC0.075 are shown in Table 1. The mass composition and the atomic composition of oxygen for PEG were found to be higher, even when the values for carbon were almost the same for PEG and CNC0.075.

The FTIR and EDX results indicate that both CNC0.075 and PEG have similar effects on the membrane in terms of functional groups, elemental mass, and atomic compositions. However, it should be noted that 2 wt% of PEG was added to the same composition of polymer solution to which 0.075 wt% of CNCs was added. Both are strong carriers of the −OH group, as shown in Figure 3. −OH group is supposed to serve as efficient hydrophilic sites, facilitating the improved roughness and better water affinity of composite membranes when they are blended with ultrafiltration membrane polymer [30].

Figure 4 shows the contact angles for both membranes containing PEG and CNC0.075. The contact angle for the membrane with PEG is slightly lower than that of CNC0.075. This means that both 2.0 wt% PEG and 0.075 wt% CNCs have similar effects on the hydrophilicity of the UF membrane. This could be attributed to the presence of strong O-H bending in both membranes. CNCs are hydrophilic biopolymers with high surface concentrations of hydroxyl (−OH). PEG is highly hydrophilic as well. It is reported in the literature that the contact angle of the UF membrane reduces with an increase in PEG [31] and an increase in CNCs [32]. However, the fact that only a small percentage of CNCs was used is remarkable. This indicates that with a relatively small amount of CNCs, the hydrophilicity of a UF membrane can be significantly improved.

SEM images for the surface at 600 nm for both membranes are shown in Figure 5. The figure shows that there is a significant difference in the structure and morphologies of the two membranes. The CNC0.075 membrane surface was relatively smooth with visible pores; this may be due to the rheological properties and phase separation behavior of the cast solution [15].

The SEM images were further analyzed using WSxM 5.0 Develop 9.1 software. A 3D image was generated, and data were obtained for pore size distribution and roughness. The 3D images for CNC0.075 reveal a smooth structure, while PEG was rough as shown in Figure 6. Figure 7 shows the surface roughness analysis for the two membranes. The root mean square roughness (RMS) for CNC0.075 was 0.82 nm, while the value for PEG was 1.323. All roughness parameters were higher for PEG than for CNC0.075. Rough membranes are known to have higher water permeability because they have a large number of water absorption sites. The pore size distribution (Figure 8) shows higher pore diameter and more pores in the PEG membrane. This may be due to the pore-forming influence of PEG itself, although both PEG and CNCs displayed good pore-forming abilities.

### 3.2. Comparing the Membrane Performance

The effect of 2 wt% PEG and 0.075 wt% CNCs on the PES/UF membrane performance was studied and compared in terms of membrane performance. The PEG additive has been known to improve membrane characteristics in terms of pore size, water flux, rejection of protein, and turbidity removal [33]. In this work, synthetic turbid water and raw restaurant wastewater were treated using CNC0.075 and PEG-blended polyethersulfone UF membranes. The initial turbidity of restaurant wastewater used was 120 NTU. The membrane performance was evaluated in terms of water flux, turbidity removal, and COD removal.

#### 3.2.1. Performance with Synthetic Water

Figure 9 shows the percentage turbidity reduction when CNCs and PEG with a concentration of 0.075 wt% and 2.0 wt%, respectively, were used in treating synthetic water with initial turbidity varying from 80 NTU to 120 NTU. The turbidity removal was higher for PEG than CNC0.075. This observation contrasts the observed turbidity removal when raw restaurant water was treated, where CNC0.075 gave higher turbidity removal. PEG is known to have a binding affinity for zeolite [34,35]. It is likely that the separation mechanism was aided by this affinity, which causes adsorptive filtration apart from restriction through the membrane pores [36]. The turbidity removal in the synthetic water was observed to increase for the two membranes from 80 NTU to 120 NTU. However, a drop in the removal was observed for turbidity of 140 NTU. Similarly, water flux reduced from turbidity of 80 NTU to 120 NTU and suddenly increased from turbidity of 120 NTU to 140 NTU (Figure 10). Generally, an increase in water flux can lead to a reduction in contaminant removal and vice versa. Water flux and contaminant removal are determined by the membrane pore size and the hydrophilicity of the UF membrane. The hydrophilicity of the membrane is an intrinsic property and cannot suddenly change. Although fouling was not studied in this work, it may be the reason for the observation. Further work is needed to determine if the pore of the membranes was affected when synthetic water of 140 NTU was treated, leading to a reduction in the turbidity removal and increase in the water flux. Nevertheless, the aim of this work to show that CNCs can be a suitable replacement of PEG in UF membrane fabrication is obvious from the results.

#### 3.2.2. Performance with Raw Restaurant Water

Similar experiments as in the previous section were conducted to evaluate the performance of the membranes when real restaurant wastewater was treated. The experiment was conducted by filtrating the waste restaurant water to find the permeability of the CNC0.075 and PEG membranes by varied pressure of 0.4, 0.6, 0.8, and 1 bar. Figure 11 shows the restaurant water flux of CNC0.075 and PEG.

Figure 11 shows the water flux of CNC0.075 and PEG UF membranes’ performance against varied pressure. The results indicate that an increase in applied pressure significantly increases the water flux of CN0.075 and PEG membranes. Based on the literature review, transmembrane pressure increases water flux because of the driving force applied [37]. The results also show that the two membranes have a similar performance in water flux, even at varied pressure. This result shows clear evidence that the CNCs and PEG are the pore-forming agents, and their presence in the polymer solution enhances the formation of more pores on the membrane surface [33,38]. The COD removal of the CNC0.075 and PEG membranes was examined as one of the basic metrics to determine the percentage removal of the two membranes in treating restaurant wastewater. The COD feed concentration used was 545.25 mg/L. Figure 12 shows the COD removal of the CNC0.075 and PEG membrane, indicating the effect of CNC0.075 and PEG on COD removal. Although the results obtained from CNC0.075 were similar to PEG to some extent, CNC0.075 had more COD removal. For instance, the highest removal percentage of the two membranes was achieved at a pressure of 0.4 bar. CNC0.075 shows the highest percentage of COD removal of 80% at 0.4 bar and lowest at 1 bar, while the PEG membrane had the highest percentage of COD removal of 60% at 0.4 bar and lowest at 1 bar. Based on the test results, CNC0.075 gave the best COD removal result. The result demonstrates that the COD removal efficiency of the membranes depends on the pore size of the fabricated membrane. This is because CNC0.075 has a lower pore size and pore size distribution.

The CNC0.075 and PEG-modified membranes were evaluated in terms of turbidity removal. Figure 13 shows the evaluation results of the two membranes as a function of percentage turbidity removal against pressure when real restaurant wastewater was treated. The figure result shows that when CNC0.075 was used at a pressure of 0.4 bar, the percentage of turbidity removal was 99.0%. As the pressure increased to 0.6 bar, the percentage of turbidity removal decreased to 98.0%. For further increases in the applied pressure to 0.8 and 1 bar, the percentage turbidity removal increased again to 99.0% and then decreased dramatically to 9.4% at 1 bar. A similar result was observed when the PEG membrane was used for treating restaurant water at varied pressure of 0.4, 0.6, 0.8, and 1 bar. Figure 13 shows that the PEG membrane produces 97.0% turbidity removal at a pressure of 4 bar. When the pressure was increased to 0.6 bar, the turbidity removal decreased to 95.0%. For further increases in pressure to 0.8 and 1 bar, the turbidity removal increased to 95.5%. These results demonstrate that increased transmembrane pressure during filtration affects turbidity removal. As a result of the pressure increase, the smaller particles can more easily pass through the membrane pore, which could lead to an increase in the turbidity level of the filtered water. Another explanation is that the presence of oil deposits in restaurant wastewater that is smaller than the membrane pore was forced into the pore by the pressure increase. The result shows that CNC0.075 has a higher percentage of turbidity removal over the PEG membrane. This observation may support the fact that higher turbidity removal was observed in the synthetic turbid water with PEG due to the interaction of PEG with zeolite. The observed better performance of CNC0.075 may be because the CNC0.075 blended well in the polymer solution, leading to a better morphology, pore structure, and hydrophilicity.

## 4. Conclusions

The PES UF membrane containing 0.075% CNCs was compared with the PES UF membrane containing 2 wt% PEG4000. The two membranes were compared based on their characteristics and performances when synthetic and raw restaurant water were treated. CNC0.075 was observed to blend well in the polymer solution, leading to better morphology, pore structure, and hydrophilicity. The water flux pattern was similar for both membranes when synthetic turbid water and real restaurant wastewater were treated. Turbidity removal was higher with PEG for the synthetic turbid water, but the reverse was the case for the real restaurant wastewater. The mechanism of separation might have been enhanced through adsorptive filtration due to the interaction of PEG with the zeolite in the synthetic turbid water. The FTIR and EDX results indicated that both CNC and PEG have similar effects on the membrane in terms of functional groups, elemental mass, and atomic compositions. However, a higher weight percent was used for PEG than CNC0.075. The contact angle for PEG was found to be almost the same as that of CNC0.075. The SEM images showed that there was a significant difference in the structures and morphologies of the two membranes. The CNC0.075 membrane surface was relatively smooth with visible pores. WSxM 5.0 Develop 9.1 software was used to perform additional analysis on the SEM images. The analysis showed that the PEG also had visible pores. A 3D picture was produced that allowed for the analysis of the pore size distribution and roughness of the membranes. While PEG’s structure was rough, CNC0.075 displayed a relatively smooth structure. The membranes revealed that both PEG and CNCs have good pore-forming influences. CNCs proved to be a suitable substitute for PEG as an additive in PES ultrafiltration membrane filtration.

## Figures and Tables

**Figure 1 polymers-15-02636-f001:**
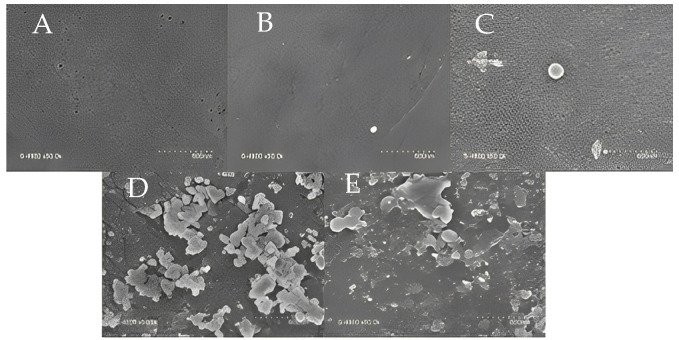
SEM images at 600 nm of PES (**A**), CNC0.05 (**B**), CNC0.075 (**C**), CNC0.1 (**D**), and CNC0.15 (**E**).

**Figure 2 polymers-15-02636-f002:**
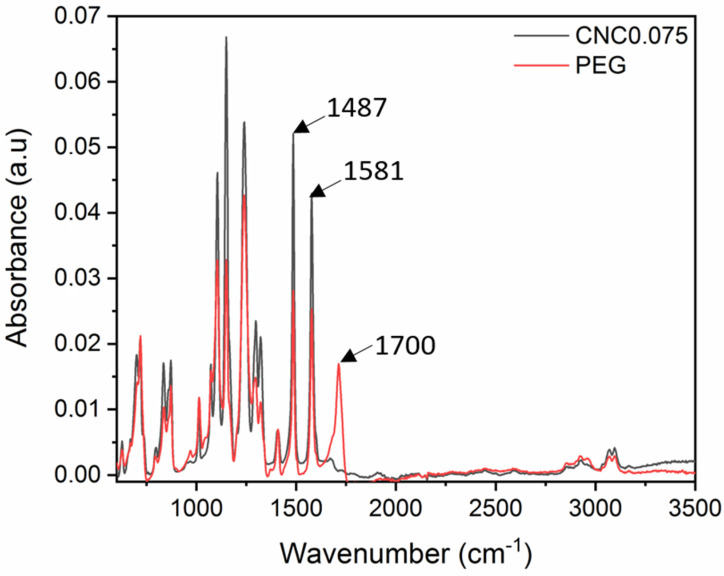
FTIR for PEG and CNC0.075.

**Figure 3 polymers-15-02636-f003:**
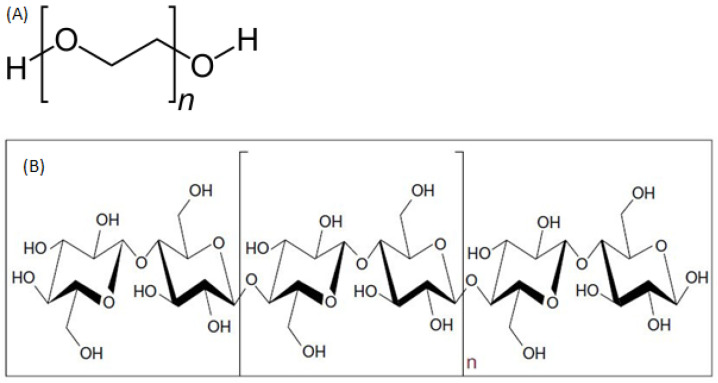
Chemical structures of PEG (**A**) and CNCs (**B**).

**Figure 4 polymers-15-02636-f004:**
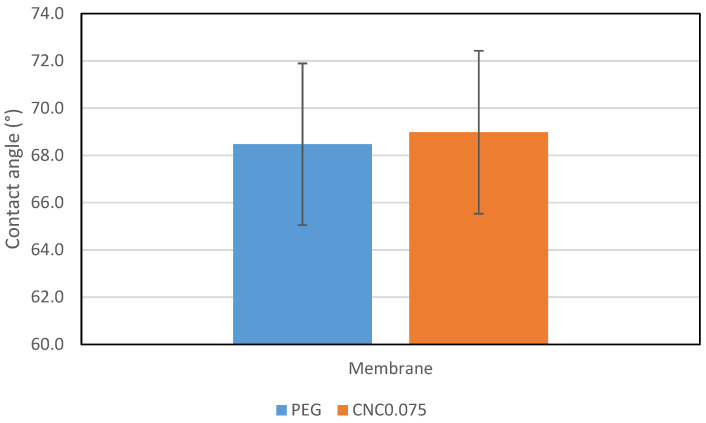
Contact angles for the UF membranes containing 2 wt% PEG and 0.075 wt% CNCs.

**Figure 5 polymers-15-02636-f005:**
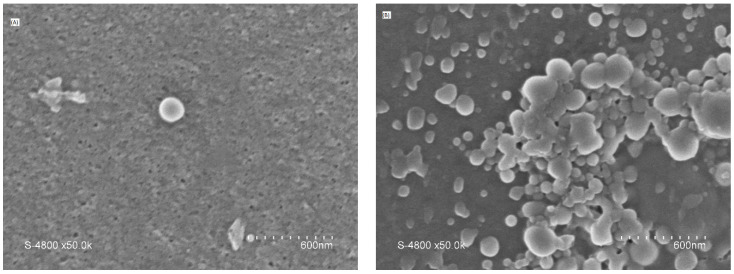
SEM images at 600 nm for CNC0.075 (**A**) and PEG (**B**).

**Figure 6 polymers-15-02636-f006:**
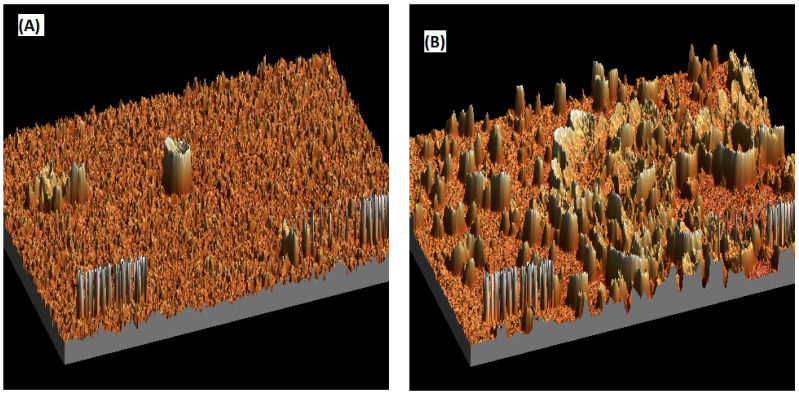
3D pictures of the SEM images at 600 nm for CNC0.075 (**A**) and PEG (**B**).

**Figure 7 polymers-15-02636-f007:**
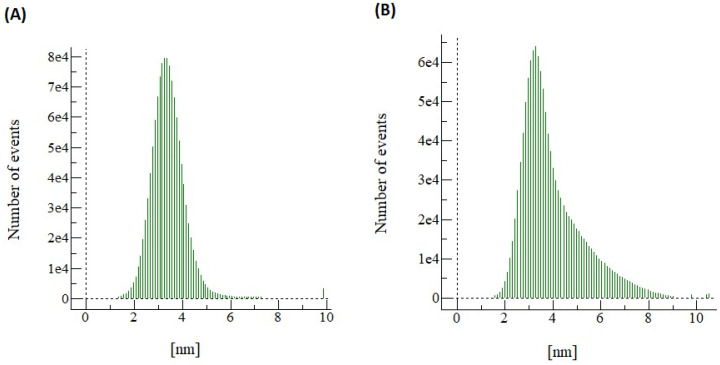
Surface roughness for CNC0.075 (**A**) and PEG (**B**).

**Figure 8 polymers-15-02636-f008:**
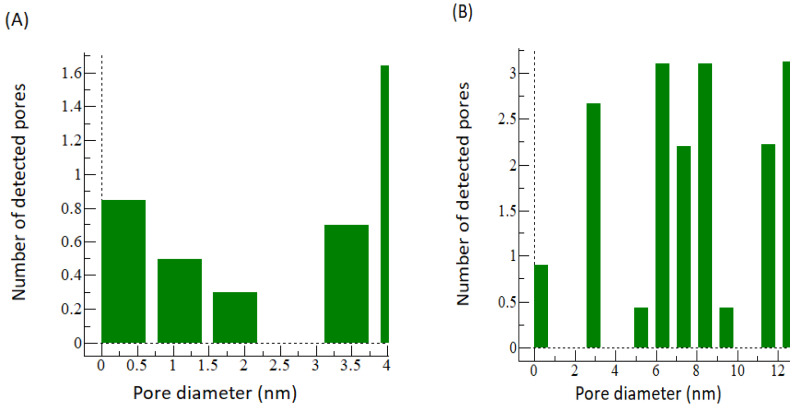
Pore size distribution for CNC0.075 (**A**) and PEG (**B**).

**Figure 9 polymers-15-02636-f009:**
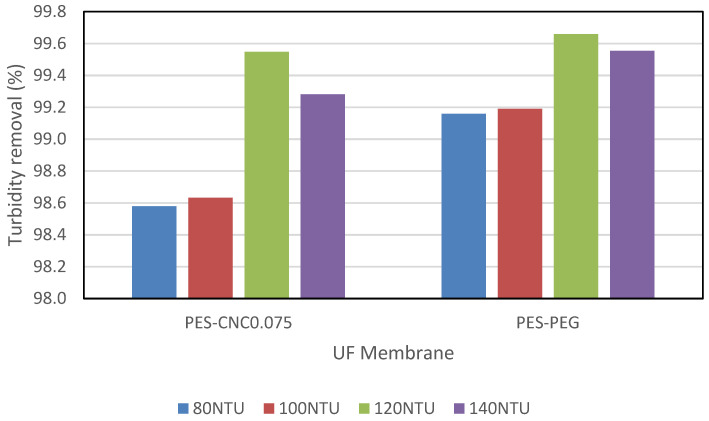
Comparing turbidity removal for PES-CNC0.075 and PES-PEG in synthetic water.

**Figure 10 polymers-15-02636-f010:**
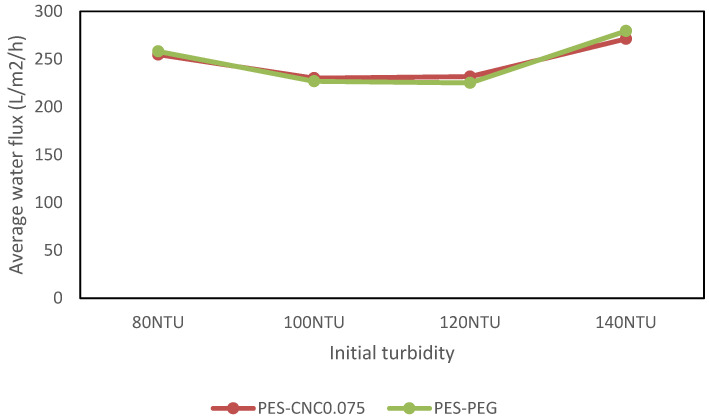
Comparing average water flux for different initial turbidity.

**Figure 11 polymers-15-02636-f011:**
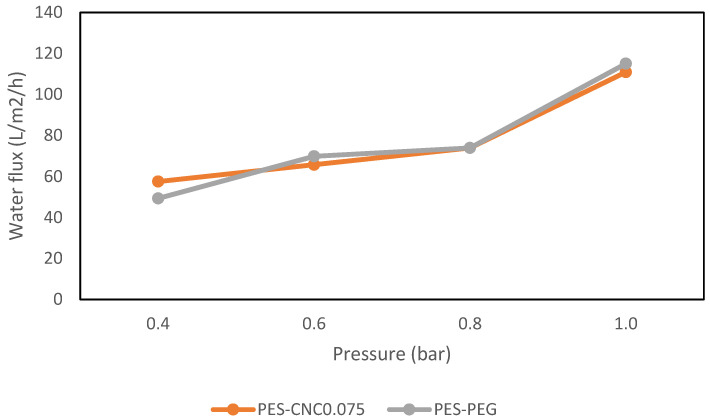
Comparing water flux for raw restaurant wastewater.

**Figure 12 polymers-15-02636-f012:**
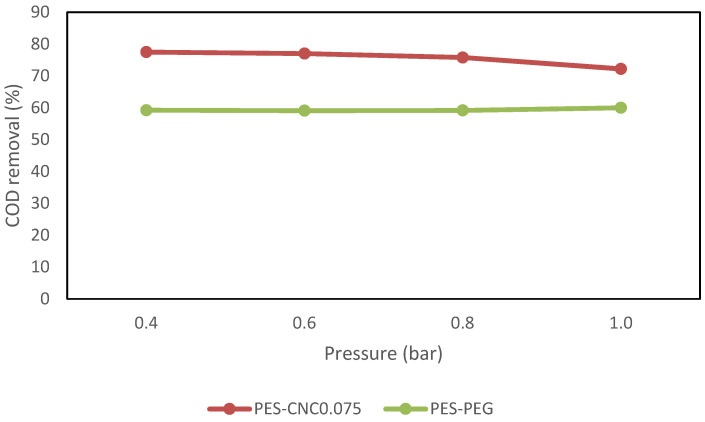
Comparing COD removal for raw restaurant wastewater.

**Figure 13 polymers-15-02636-f013:**
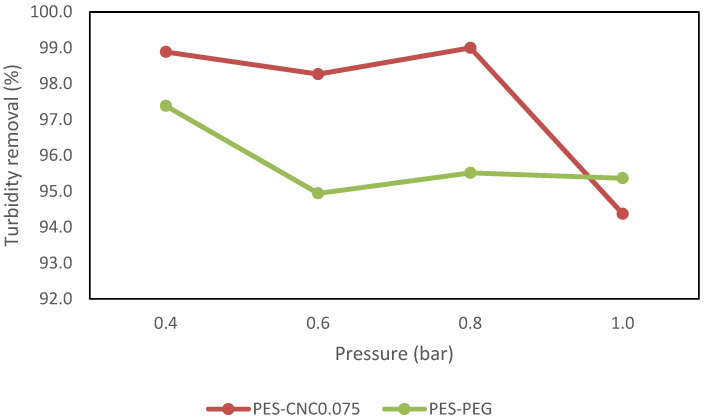
Comparing turbidity removal for restaurant wastewater.

**Table 1 polymers-15-02636-t001:** EDX results for PEG and CNC0.075.

	Mass Composition		Atomic Composition	
	PEG	CNC0.075	PEG	CNC0.075
Carbon	68.79%	68.71%	77.04%	77.81%
Oxygen	23.45%	19.72%	20.92%	17.78%

## Data Availability

Not applicable.

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
