# Peer review of "A Comparison of the Effect of Cellulose Nanocrystals (CNCs) and Polyethylene Glycol (PEG) as Additives in Ultrafiltration Membranes (PES-UF): Characterization and Performance"

_polymers, 2023, doi:10.3390/polym15122636_

Round 1

Reviewer 1 Report

1. Line 21 – sentence is not finished.

2. Line 31 – missing article

3. The Introduction should be significantly improved in terms of it’s supplement to the work itself. 

4. Some lines are not related to the work, e.g. Line 34: “Domestic wastewater includes yellow water, black water, brown water, and grey water”. This is not a relevant classification for this work, and not a very common. The authors could relate this to the work, by adding what color is the studied restaurant water. Personally I have only seen brown or yellow water in domestic wastewater.

5. Line 50: sentence with poor grammar

6. Line 64: I would argue with the authors that the operating pressure for UF is 200-700 kPa. Firstly, it should be stated that it’s the TRANSmembrane pressure, secondly he normal operating pressure very rarely exceeds 1 bar, with 1 bar being a rather critical value, 2 bars being the maximum operating value.

7. Line 67-69: “The type of contaminant UF can remove is limited mainly by the porosity and hydrophobicity of the membrane. Therefore, the use of naturally synthesized additives could improve the characteristics and, hence, the performance of the membrane” The type of contaminant is limited by the pore size and the contaminant size in UF, obviously not by porosity or hydrophobicity. Not clear how in this context the use of naturally synthesized additive can improve the characteristics.

8. Materials section is composed poorly. No features of the materials is presented. E.g. it’s very crucial for this work to give the size of Cellulose nanocrystals and it’s Mw.

9. Line 106: “where the solvent rich liquid phase also known as polymer-lean phase, will originate from the membrane pores.” Wrong: pores originate from the polymer lean-phase, not vice versa.

10. Methods section is very poorly described. The polymer solution 18% wt. cannot be stirred using a magnetic stirrer (line 109). What was the solution temperature?

11. Authors should provide the designation (name) of the equipment  and properly described the sample preparation.

12. Formulas are presented not in accordance with the journal rules.

13. Instead of fig. 2 and lines 151-164 the authors could provide the table, which would give more information to the reader.

14. Word “Figure” is reduced to “Fig” in scientific articles

15. Fig. 4: the authors should provide error bars. To me the contact angle is the same. Besides such information should be given in the overall comparison table. Fig. 4 looks poor.

16. Line 190:  why do the authors state PEG was observed on the surface? Maybe the other sample would look similar if the micrograph was taken on the other place. 

17. Fig. 6 – scale bars should be given. The data should be supported by ASM measurements.

18. Fig. 8 – not all data is provided, obviously there are more pores to the right of each graph.

19. Line 213 – “more pores in the PEG membrane” vs line 190: “The PEG has no visible pores”

20. Line 229: “The turbidity removal was higher for PEG than CNC0.075. This may be due to the binding affinity of zeolite in the synthetic turbid water to PEG rather than the membrane separation efficiency” How do the authors support this thesis? What is the logic behind? What is the concentration, size and the type of zeolite used?

21. Why do lines 252 – 260 describe each point on Fig. 11.? A table could be provided instead. It’s a common knowledge that the flux rises with TMP, since it’s inverse proportionality. What does the fig. 11 convey, why providing it as a fig?

22. In the results and discussion, the authors don’t place any logic behind explanation. A lot of their conclusions in this section is arguable. 

The major concern of this work is that its basis is comparison of 0,075% wt of CNC with 2% PEG. The produced membranes have little to no difference, because the amount of additive is too small. To support their thesis for CNC the authors should provide measurements of consecutive increase of CNC (larger amounts) to see if there is any difference and compare it to 0% CNC. The authors support their work by citing themselves [21], but I was not able to find this work in the open sources.

Must be improved

Author Response

All comments have been addressed.

Reviewer 2 Report

The manuscript is on an interesting topic and the results can be useful. However, it is very difficult to read. My suggestion is for the authors to work hard on the writing and resubmit. Thanks.

Cannot meet the basic scientific writing std.

Author Response

The comment has been addressed.

Reviewer 3 Report

 Adeniyi et al prepared two polymer membranes by incorporating CNC and PEG with DES which showed good performance for the treatment of restaurant wastewater. The CNC membrane showed higher turbidity and COD removal capability. The findings could be of interest for researchers working on water purification. This manuscript can be published after the concerns below being addressed.

1. The percentage of PEG and CNC in this work are 2wt% and 0.075% respectively which is a significant difference. What is this selection of percentage based on? A brief clarification will be meaningful for the readers.

2. What is the reason for the drop of turbidity removal in 140 NTU for both membranes? It is also apparently seen that there is a flux increment in 140 NTU for both membranes? The authors need to comment on these observations. Additionally, are there any correlation between the two phenomenon?

3. Line 230, the authors need to be more specific for the binding affinity between zeolite and PEG in terms of molecular interactions between PEG and zeolites.

4. What is the pressure for the test with synthetic turbid water?

5. More details about how the 3D image and pore size analysis results were generated need to be provided.

6. The concept of ‘restaurant wastewater’ is very broad which may not accurately reflect what polluting components the membrane is capable of removing. A brief description of the major pollutants in the restaurant wastewater needs to be included.

7. The authors need to define the abbreviation COD.

Author Response

All comments have been addressed.

Round 2

Reviewer 1 Report

Dear authors, thank you for considering my comments.

I see that the authors have attended to some of my comments and have made changes. 

However, the major drawbacks of the work were not attended to, namely: 

1. The major concern of this work is that its basis is comparison of 0,075% wt of CNC with 2% PEG. The produced membranes have little to no difference, because the amount of additive is too small. To support their thesis for CNC the authors should provide measurements of consecutive increase of CNC (larger amounts) to see if there is any difference and compare it to 0% CNC. 

2. The choice of test methods is arguable.

3. The description of materials, methods, and techniques is poor.

4. Poor presentation and interpretation of the data obtained in the results and discussion section.

Given that the most significant drawbacks were not attended I can’t change my opinion.

The decision is up to the academic editor.

 Moderate editing of English language required